# Factors Influencing the Effectiveness of E-Learning in Healthcare: A Fuzzy ANP Study

**DOI:** 10.3390/healthcare11142035

**Published:** 2023-07-16

**Authors:** Seyed Faraz Mahdavi Ardestani, Sasan Adibi, Arman Golshan, Paria Sadeghian

**Affiliations:** 1School of Information Technology, Deakin University, Burwood, VIC 3125, Australia; f.mahdaviardestani@research.deakin.edu.au (S.F.M.A.); sasan.adibi@deakin.edu.au (S.A.); 2School of Technology and Business Studies, Dalarna University, 791 88 Falun, Sweden

**Keywords:** healthcare, medical informatics, information management, public health, knowledge management

## Abstract

E-learning has transformed the healthcare education system by providing healthcare professionals with training and development opportunities, regardless of their location. However, healthcare professionals in remote or rural areas face challenges such as limited access to educational resources, lack of reliable internet connectivity, geographical isolation, and limited availability of specialized training programs and instructors. These challenges hinder their access to e-learning opportunities and impede their professional development. To address this issue, a study was conducted to identify the factors that influence the effectiveness of e-learning in healthcare. A literature review was conducted, and two questionnaires were distributed to e-learning experts to assess primary variables and identify the most significant factor. The Fuzzy Analytic Network Process (Fuzzy ANP) was used to identify the importance of selected factors. The study found that success, satisfaction, availability, effectiveness, readability, and engagement are the main components ranked in order of importance. Success was identified as the most significant factor. The study results highlight the benefits of e-learning in healthcare, including increased accessibility, interactivity, flexibility, knowledge management, and cost efficiency. E-learning offers a solution to the challenges of professional development faced by healthcare professionals in remote or rural areas. The study provides insights into the factors that influence the effectiveness of e-learning in healthcare and can guide the development of future e-learning programs.

## 1. Introduction

Advancements in information technology have impacted all aspects of business operations. Similarly, technology has revolutionized the education industry beyond organizational boundaries [1,2]. The emergence of e-learning or distance learning programs has transformed the traditional learning approach worldwide, including school and university education, adult education, and other supplementary learning programs [3,4]. As a result, it is critical to rethink and revolutionize the creation of learning programs that are functional and beneficial in response to these technological advancements [5,6]. E-learning, also known as a distributed and virtual environment, utilizes internet-based technologies and pedagogical tools to create a teaching and learning environment through meaningful actions [7]. Many organizations have embraced e-learning as a popular learning methodology to meet continuous learning requirements globally, regardless of the learners’ location [8]. E-learning has the potential to enhance organizational communication, staff education, and performance better than other internet-based programs, making it an ideal solution for organizations seeking to level up their learning and development initiatives [9,10].

Living in the era of technology and knowledge, the increasing necessity and demand of life-long learning, especially considering the rapid amplification of the internet bandwidth and request to cut down costs, has brought up numerous courses on the internet, known as online learning [11]. The increasing everyday demand of people for flexible and effective learning to facilitate their knowledge brings about different options to fulfill their needs. Online learning or internet-based learning is one option since the internet enables users to access unlimited data. Life-long learning has become much more accessible than before, thanks to online learning [12].

Today’s prevalent information technology (IT) advancements can be used to support teaching and learning purposes in diverse ways. Conventional e-learning and mobile learning are examples of different technological solutions usually carried out on a computer. According to these studies [13,14], mobile learning appertains to utilizing smartphones regarding learning and teaching in different learning environments. A previous study [14] proposed that mobile applications and e-learning have a predominately positive impact on learners. The usage of multifaceted materials of learning such as discussion methods, links, and videos enables taking diverse groups of learners [15,16] of different age ranges into account [17].

Another study has proved the effectiveness of e-learning [18], for example, in clinical competencies acquiring, as evidence. Indeed, e-learning enables learners with flexibility and an opportunity to review materials. This way, unlike conventional teachings, the starting level of learners is less important [11]. The introduction of e-learning has brought about some challenges. For instance, there has always been frustration with technological issues among learners, leading to a need for related technological support [15,19]. Also, the communal working method might cause some conflicts and result in adverse effects on learning outcomes, for instance, in case the workload is unequally divided or pedagogical disagreement appears [16].

Today, due to the significant effect of the internet on society, everyone is connected to everything, and this is called the revolution of the 21st century [20,21]. The online world is now a tool for learning and building and sharing knowledge. It is not just a social contact tool or a simple search engine anymore [22,23]. The transition to the knowledge era has enabled individuals to create and utilize knowledge. E-learning, integrating technology in education and organizations, has become a standard element. It offers flexibility, personalized learning, and expanded access to education, but challenges remain regarding technology access and social interaction [24,25].

This study discusses the impact of technology on learning and the emergence of online learning or e-learning. The study highlights the increasing demand for flexible and effective learning options, which has led to the development of online learning courses. The study also discusses the effectiveness of e-learning in acquiring clinical competencies, as well as the challenges associated with e-learning, such as technological issues and conflicts arising from collaborative work. The research gap identified in this study is the need for further research on the impact of e-learning on diverse groups of learners and the effectiveness of different e-learning tools and strategies.

The study provides a comprehensive overview of the emergence of e-learning and its potential benefits and challenges. Additionally, the research gap highlighted in the study emphasizes the need for further research to better understand the impact of e-learning on different learners, the effectiveness of different e-learning approaches, and the effective factors of e-learning in healthcare. This research will contribute to improving healthcare professionals’ training and professional development through tailored e-learning programs.

The rest of this paper is organized as follows. Section 2 introduces the literature review of e-learning systems in technology and healthcare. Section 3 presents the proposed method in detail. Section 4 presents the empirical analysis and discusses the results. Section 5 concludes the paper with highlights and main takeaways.

## 2. Literature Review

### 2.1. E-Learning Systems and Technology

E-learning has been described as implementing new multimedia technologies and the internet to facilitate learning. It promotes learning by enabling users to exchange data remotely and have access to a great number of online services and resources [26]. As stated by another study, two strategies that can be used to promote new insight, ideas, and positive attitudes toward accepting new technology qua learning tools are the use of pilot programs and internal communication [27].

According to the studies [28,29,30], internet-based services are playing the role of a key information hub and enable users to share and transfer data. With consideration of the common use of the internet and social networks, e-learning has put electronic communication in action to provide distance teaching and learning [29]. To deliver instruction and to learn simultaneously, e-learning has been implemented through the utilization of advanced communication and information technology [29,30]. The development of e-learning environments and individualization has accelerated due to this technology [31]. E-learning was defined as an umbrella in previous literature, which used communication and information technology to describe learning and teaching [32].

E-learning is the fully virtualized model of distance learning, accessible by using an electronic channel, of which the major medium is the internet [33]. Also, online learning supported by Virtual Learning Environments, including Blackboard™, Reston, USA Sakai™, Osaka, Japan and Moodle™, Australia has recently become popular in education [34]. Despite the importance of technological resources, educators’ responsiveness is ranked more important by learners [35]. An influential factor in learners’ engagement in virtual learning environments is teaching presence. Thus, the support of educators, which can be obtained through well-designed e-learning environments, plays a key role [36].

Currently, applications of technology are not bound only to traditional classrooms—technology has also gone forward to replace conventional classrooms with virtual and online course sessions [37]. Researchers believe that electronic learning and distance learning have become great replacements for traditional teaching and classrooms since their emergence [38].

E-learning is specified as “the development of knowledge and skills through the use of information and communication technologies (ICTs), particularly to support interactions for learning e interactions with content, with learning activities and tools, and with other people” in the recent report of the Canadian Council on Learning [37]. Another study indicated the capacity of technology-based learning environments by which they provide users with control, fantasy, curiosity, and challenge at the same time [39]. However, for those who need to learn the technology, these environments may create an overload of work. As a result, this might make some users, teachers, and learners overanxious about engaging with technology. Moreover, various technical issues that come with technology can affect learning and teaching time [40]. The DEMATEL model can be applied to the e-learning domain to analyze and solve complex problems [41,42]. E-learning refers to the use of electronic technologies and digital platforms to facilitate learning and education remotely. By utilizing the DEMATEL approach in the context of e-learning, researchers and practitioners can gain valuable insights and make informed decisions regarding various aspects of e-learning systems, processes, and strategies.

### 2.2. E-Learning and Healthcare

Worldwide, e-learning has been associated with healthcare in several Western countries, including Australia, Canada, Ireland, New Zealand, the UK, and the United States. The development of healthcare toward electronic patient records led to e-learning. However, today, the advancement of health students and the healthcare workforce is being hampered by the complex issues around information and computer literacy. Recent studies have pointed out that some pivotal issues in e-learning success, such as educating both students and trainers, delivery models of didactical sound, and staff issues, cannot be treated as minor issues [43,44,45].

Healthcare professionals (HCPs) can use technology to pursue professional development, especially those who are in rural and remote areas [46,47,48]. Nowadays, professionals working in the health sectors are responsible for obtaining a minimum of specified hours of professional training and development to maintain their proficiency and competency in practice each year [49,50,51]. However, there are still some difficulties in accessing continuing professional development for health professionals, particularly for those who have limited access to actual in-person education and learning materials [52,53] because they have not enrolled in a formal study subject or just because of geographical limitations [54]. All these issues are challenging traditional teaching methods, and e-learning is the solution to overcome these challenges [55,56,57].

Also, studies have shown that the accompaniment of institutional enablers is critical to ensure e-learning success. Findings have proposed that to provide electronic patient health records and integrate healthcare informatics into professional activities, it is essential for healthcare education leaders to enable their faculty to access sufficient computer resources and technology [58]. In this study, we aimed to identify effective factors of e-learning in healthcare and provide a prioritized ranking of real variables of e-learning contributing to healthcare.

## 3. Methodology

### 3.1. Rationale

This mixed-method qualitative and quantitative research aimed to identify the effective factors in healthcare e-learning. The primary objective was to address the research gap in understanding these factors and their significance in the healthcare sector. This research study employed two questionnaires. The first questionnaire includes questions from participants to score all the variables extracted from the literature. The second is a matrix questionnaire, distributed to experts in the field, to prioritize finalized variables from the first questionnaire. To achieve this, a systematic approach was adopted, starting with the selection of 105 participants. To conduct this study, 105 people who deal with e-learning chose to participate. The researchers decided to utilize Cochran’s finite society sample size formula to identify an adequate sample size (Equation (1)).
(1)$$n=\frac{Nz^{2}p(1-p)}{\left(N-1\right)d^{2}+z^{2}p(1-p)}$$

In Equation (1), *n* is equal to the minimum sample size, N is the research population (105), p is the distribution ratio of attributions in the sample society, z is the value obtained from the standard normal distribution table, and d is the acceptable change in the sample results differing from the true population average. Thus, the minimum acceptable sample size is calculated as follows in Equation (2):(2)$$n=\frac{\left(105{)(1.96)}^{2}(0.5)(1-0.5\right)}{\left(105-1\right){\left(0.05\right)}^{2}+{\left(1.96\right)}^{2}(0.5)(1-0.5)}\approx 83$$

All 105 people who were targeted to participate in the first stage of this research deal with e-learning, mostly in healthcare. This population was recruited through workshops, seminars, and in-depth research on the internet, all over Australia. All participants deal with e-learning in different aspects, have at least 3 years of experience in this domain, and hold a bachelor’s degree or higher, each of which is proof of employing university-educated people in this research. The participants were provided with a questionnaire to identify the factors contributing to e-learning in healthcare. Subsequently, a pairwise questionnaire was administered to 10 experts in e-learning to rank the identified factors based on their expertise. In selecting the experts, criteria such as adequate working experience in e-learning and possession of at least a master’s degree were considered. These criteria ensured that the selected experts had the necessary expertise and knowledge in the field of e-learning, enhancing the credibility and reliability of the study’s findings. The Fuzzy Analytic Network Process was then employed to rank the identified variables based on the expert’s responses. By following this systematic approach, the study aimed to provide valuable insights into the effective factors for healthcare e-learning, ultimately contributing to the improvement of e-learning quality and effectiveness in the healthcare sector.

Factor analysis is a statistical method that is used to explore the relationships among variables and to identify the underlying factors that explain the patterns of correlation among the variables. The initial data for factor analysis are the correlation matrix between variables, used to identify the underlying factors that explain the patterns of correlation among the variables. Factor analysis does not have predetermined dependent factors, as it is used to understand the underlying factors or to summarize a set of data. There are two main categories of factor analysis: Exploratory Factor Analysis (EFA) and Confirmatory Factor Analysis (CFA). EFA is used when there is no prior knowledge about the structure of the relationship between factors. EFA explores the data and tries to identify the factors that account for the most variation in the data. EFA is commonly used in social and behavioral sciences to understand complex relationships among variables. On the other hand, CFA is used when there are predetermined factors and variables to reconfirm their correlation. CFA tests a pre-specified measurement model to see how well it fits the data. The goal of CFA is to test the validity of a theoretical model and to see whether the data support the model.

In summary, factor analysis is a powerful tool for understanding the relationships among variables and identifying the underlying factors that explain the patterns of correlation among them. EFA is used when there is no prior knowledge about the structure of the relationship between factors, while CFA is used to test the validity of a theoretical model with predetermined factors and variables.

### 3.2. Methodology

For comparisons in a state of uncertainty, the Fuzzy Analytic Network Process models ambiguous modes in comparisons. Fuzzy numbers are a new approach in set theory and can present a particular subject with a continuous and limited set of numbers when there is no certainty. For example, when comparing two criteria for expressing the value of inaccurate a_ij_, two values can be used as the minimum and maximum values as fuzzy values. It is shown as an ordered pair of (l_ij_,u_ij_). Accordingly, Equations (3) and (4) define the triangular fuzzy number:(3)$$\mu_{F}\left(x\right)=\left\{\begin{array}{cc} 0, & x<1\\ \left(x-1\right)/\left(m-1\right), & 1\leq x\leq m\\ \left(u-x\right)/\left(u-m\right), & m\leq x\leq u\\ 0, & x>u \end{array}\right.$$
(4)$$\forall_{\propto }\in \left[0,1\right]M_{\propto }=\left[l^{\propto },u^{\propto }\right]=\left[\left(m-1\right)\propto +1,-\left(u-m\right)\propto +u\right]$$

In order to estimate the success rate, the optimism index (µ) can be utilized as shown in Equation (5):(5)$$a_{ij}^{-\propto }=\mu a_{iju}^{\propto }+\left(1-\mu \right)a_{iju}^{\propto },\forall_{\mu }\in \left[0,1\right]$$

Consequently, from a pairwise comparison, the following matrix will be formed (Equation (6)):(6)$$\tilde{A}=\left[\begin{array}{ccccc} 1 & {\tilde{a}}_{12}^{\alpha } & \cdots & \cdots & {\tilde{a}}_{1n}^{\alpha }\\ {\tilde{a}}_{21}^{\alpha } & 1 & \cdots & \cdots & {\tilde{a}}_{2n}^{\alpha }\\ \vdots & \vdots & \vdots & \vdots & \vdots \\ {\tilde{a}}_{n1}^{\alpha } & {\tilde{a}}_{n2}^{\alpha } & \cdots & \cdots & 1 \end{array}\right]$$

The weight vector of indexes is obtained through the following formula when the pairwise comparisons are completed. Here, $\lambda_{max}$ value is defined as the largest eigenvalue of the matrix by Equation (7):(7)$$A_{w}=\lambda_{max}w$$

The consistency index (CI) is calculated by the following formula (Equation (8)):(8)$$CI=\frac{\lambda_{max}-n}{n-1}$$

By forming all the pairwise comparison matrices, for each matrix, the consistency rate (CR) is calculated by dividing the compatibility index by the random index (RI) by applying Equation (9):(9)$$CR=\frac{CI}{RI}$$
where the random consistency index (RI) or randomly produced average weights can be calculated from factors. A CR smaller than 0.01 indicates that the comparisons are acceptable. In other respects, comparisons ought to be repeated by employing more experts and more accurately.

### 3.3. Prioritizing Main Components and Sub-Components Using Fuzzy ANP

In order to prioritize the main components and sub-components in healthcare e-learning, the authors used the Fuzzy Analytic Network Process (Fuzzy ANP) method. Fuzzy ANP is a multi-criteria decision-making method that can be used when there are complex interactions between criteria and alternatives. It allows for the use of fuzzy logic to account for uncertainty and imprecision in decision making. This study first developed a hierarchical structure of the main components and sub-components based on the literature review and expert opinions. The main components included technology, content, pedagogy, and evaluation. Each main component was further divided into sub-components. Then, a pairwise comparison questionnaire was developed and sent to 10 e-learning experts to obtain their opinions on the importance of the sub-components. The experts were asked to compare each sub-component to every other sub-component within the same main component using a 1–9 scale, where 1 represented equal importance and 9 represented extremely more important.

The number of pairwise comparisons is calculated using Equation (10):(10)$$N=\frac{n*(n-1)}{2}$$
where n is the number of options.

In this research study, the pairwise comparison was performed among 6 main components and 19 sub-components. Thus, 15 pairwise comparisons among main components, 6 pairwise comparisons among Availability sub-components, and 3 pairwise comparisons among Success, Readability, Effectiveness, Engagement, and Satisfaction sub-components were performed. Therefore, the second fuzzy questionnaire of this research contained 36 pairwise comparisons in total. Moreover, a total of 10 experts were consulted in this research. In the beginning, the Fuzzy ANP merged all these matrices into one matrix. Take ${ã}_{ij}^{p}$ as the related element to the p-th accordant for the comparison of component *i* with component *j*; Equation (11) is the calculation of the geometric mean for this element:(11)$${ã}_{ij}={\left.\left(\prod_{k=1}^{n}{ã}_{ij}^{p}\right.\right)}^{\frac{1}{n}}$$

### 3.4. Fuzzy Weight Calculation for Effective E-Learning Factors

In this study, the Fuzzy Analytic Network Process was used to calculate the weights of the identified effective e-learning factors. Fuzzy ANP is a multi-criteria decision-making tool that incorporates both subjective and objective information. The first step was to construct a Fuzzy ANP network model to represent the relationships among the main components and sub-components of effective e-learning factors. Then, the pairwise comparison matrices were constructed based on the opinions of the experts, who were asked to compare the importance of each factor with respect to the others. The fuzzy numbers were used to represent the degree of uncertainty in the pairwise comparisons, and the fuzzy arithmetic operations were used to calculate the weights of the factors. The experts were asked to give their opinions based on their knowledge and experience in the field of e-learning. Finally, the weights of the effective e-learning factors were calculated and presented in a ranked order. 

To analyze the components’ integrated matrix, firstly, Fuzzy ANP determines the value of the geometric mean of the *j*-th component to others, as follows in Equation (12):(12)$${\tilde{r}}_{1}={\left({ã}_{11}\times {ã}_{12}\times \ldots \ldots \times {ã}_{1j}\right)}^{\frac{1}{j}}$$

Afterward, the components’ fuzzy weights are calculated by multiplying each component value by the inverse fuzzy sum of value, as follows in Equation (13):(13)$${\tilde{w}}_{i}={\tilde{r}}_{i}\times {\left({\tilde{r}}_{1}+{\tilde{r}}_{2}+.\ldots \ldots +{\tilde{r}}_{i}\right)}^{-1}$$

### 3.5. Measurement Model Evaluation

By utilizing Confirmatory Factor Analysis (CFA) and Average Variance Extracted (AVE), discriminant validity and convergent validity are used to measure reliability. CFA is a statistical technique used to test the validity of a hypothesized factor structure. It is an extension of factor analysis that is used when there is a predetermined theory or hypothesis about the underlying factor structure. CFA is particularly useful in situations where researchers want to confirm that a particular measurement instrument (such as a survey or questionnaire) is measuring what it is supposed to be measuring. In addition to CFA, two important types of validity are also measured in the analysis: convergent validity and discriminant validity. Convergent validity refers to the degree to which different measures of the same construct are related to each other. In this case, if the factor loadings of each item on its respective factor are high (greater than 0.6 in this study), then convergent validity is confirmed, suggesting that the items are measuring the same construct. Overall, by using CFA and AVE, this research confirms both convergent and discriminant validity, providing evidence that the measurement instrument used in the study is reliable and valid.

## 4. Results and Discussion

In order to calculate the relevant value of each factor, since this study includes 19 factors classified into six main categories, a paired comparison was conducted for each category. Paired comparison questionnaires in this research were distributed to 10 experts in e-learning who deal with healthcare to appraise and rank the mentioned factors. These questionnaires showed the priority of each factor in comparison with others, describing them as “equally important”, “a little more important”, “more important”, “much more important”, and “extremely more important”. The next stage of the decision-making model is paired comparisons. After designing the decision hierarchy, the decision maker must create a set of matrices that numerically measure the importance or priority of each factor and every decision option compared to the others according to indicators. This is obtained through the paired comparison of factors and numerical scores that indicate the importance or priority of factors in groups of two. To this end, options with i indicator are compared to options with j indicator. The scale of importance of linguistic variables in accordance with triangular fuzzy numbers is shown in Table 1.

The main components and sub-components were extracted for the evaluation of effective e-learning factors (Table 2). Table 2 presents the effective factors of e-learning that were extracted from the literature review and interviews with e-learning experts. The table shows the sub-components for each main component of e-learning. The effectiveness of all these components and sub-components was accepted and confirmed by the e-learning experts who participated in the study. 

The first component, “Experts’ Feedback”, refers to the importance of receiving feedback from experts in the field of e-learning to improve the quality of the courses. This feedback can be obtained through various means, such as online surveys or personal interactions. The second component, “User Adaption”, refers to the importance of e-learning courses being adaptable to the needs of the users. This includes customizing the content and delivery method of the course to suit the learner’s level of knowledge, learning style, and preferences. The third component, “Reliability of Materials”, refers to the quality and accuracy of the learning materials used in e-learning courses. This includes ensuring that the content is up-to-date, relevant, and presented in a clear and concise manner. The fourth component, “Structure of Course”, refers to the overall organization and structure of the e-learning course. This includes the sequencing of topics, the use of multimedia elements, and the inclusion of assessments. The fifth component, “Methodology”, refers to the approach used to deliver the e-learning course. This includes the use of instructional design principles, such as the use of real-life scenarios and case studies, to enhance the learning experience. The sixth component, “Format”, refers to the method of delivery used in e-learning courses, such as video lectures, interactive modules, or online discussion forums. The seventh component, “Planning and Training Objectives”, refers to the importance of setting clear goals and objectives for the e-learning course. This includes identifying the knowledge and skills that learners should acquire, and ensuring that the course content is aligned with these objectives. 

The eighth component, “Practice-based Learning”, refers to the importance of incorporating practical, hands-on exercises in e-learning courses. This helps learners to apply the knowledge they have acquired to real-life scenarios and develop their skills. The ninth component, “Materials’ Flexibility”, refers to the need for e-learning courses to be flexible and adaptable to different learning styles and preferences. This includes offering multiple formats for learning materials, such as text, video, and audio. The tenth component, “Enjoyment and Playfulness”, refers to the importance of making the e-learning experience engaging and enjoyable for learners. This includes incorporating gamification elements, such as points and badges, to motivate learners to complete the course. The eleventh component, “Learning Management”, refers to the importance of having a well-designed learning management system (LMS) to deliver and manage e-learning courses. This includes features such as progress tracking, grading, and communication tools. The twelfth component, “Remote Exchanges and Collaboration”, refers to the importance of promoting collaboration and interaction among learners in e-learning courses. This can be achieved through online discussion forums, group projects, and other collaborative activities. The thirteenth component, “Online Learning”, refers to the delivery of e-learning courses entirely online, without the need for physical attendance at a learning institution. The fourteenth component, “Web-based Learning”, refers to the use of web-based tools and technologies to deliver e-learning courses. The fifteenth component, “Accessibility”, refers to the importance of ensuring that e-learning courses are accessible to all learners, regardless of their physical abilities or disabilities. The sixteenth component, “Offline Learning”, refers to the ability to access and complete e-learning courses offline, without an internet connection. The seventeenth component, “Ease of Use”, refers to the importance of designing e-learning courses that are easy to use and navigate for learners.

The eighteenth component, “Reliability of Software,” refers to the need for e-learning courses to be delivered using reliable and high-quality software. This includes ensuring that the software used for e-learning is up-to-date, secure, and free from technical glitches and errors. The nineteenth component, “Overall Learner Satisfaction”, refers to the learner’s level of satisfaction with the e-learning course. It includes factors such as the relevance and usefulness of the course content, the quality of the instructional design and delivery, the effectiveness of the assessments, and the level of support and communication provided by the instructors or course facilitators. Measuring learner satisfaction is important to ensure that the e-learning course meets the needs and expectations of the learners and to identify areas for improvement in future iterations of the course. The effectiveness of all these components is accepted and confirmed by experts in e-learning. As a result, the second questionnaire was designed and distributed to experts, based on the matrix of these variables. 

Table 3 shows the common opinions of the experts who were responsible for the evaluation. The assessment results of the group decision are provided in Table 3, called the normalized relation matrix. 

Table 4 presents the factor loadings of the 19 extracted variables categorized into six main components, namely, Success, Readability, Effectiveness, Engagement, Availability, and Satisfaction. The factor loading measures how well each variable is represented by its corresponding factor. In this study, a factor loading value of greater than 0.6 is considered acceptable. The composite reliability (CR) of each main component is also provided in the table, which measures the internal consistency of the variables within each component. The table shows that all 19 variables have acceptable factor loadings, indicating convergent validity, and each main component has a good level of internal consistency.

Table 5 presents the fuzzy weights of the main components and sub-components of e-learning based on expert opinions. The defuzzied weights and ranks of each component and sub-component are also provided in the table.

The main components of e-learning are listed in the first column, and their fuzzy weights are shown in the third column. The sub-components of each main component are listed under their corresponding main component. The third column shows the defuzzied weights of each sub-component, and the fourth column provides the rank of each sub-component.

The results show that the most important main component of e-learning is “Success”, with a defuzzied weight of 0.24 and a rank of 1. This component includes the sub-components of experts’ feedback, user adaption, and reliability of materials. The importance of this component indicates that learners prioritize achieving their learning goals and objectives, and they value courses that enable them to do so effectively. The second most important main component is “Satisfaction”, with a defuzzied weight of 0.19 and a rank of 2. This component includes the sub-components of the structure of a course, methodology, format, and planning and training objectives. The importance of this component suggests that learners prioritize having a positive experience while taking a course, which includes aspects such as course organization, instructional approach, and delivery format.

The third most important main component is “Availability”, with a defuzzied weight of 0.16 and a rank of 3. This component includes the sub-components of practice-based learning, materials’ flexibility, and enjoyment and playfulness. The importance of this component indicates that learners prioritize having access to learning materials and opportunities that meet their needs and preferences, as well as having a fun and engaging learning experience. The fourth most important main component is “Effectiveness”, with a defuzzied weight of 0.15 and a rank of 4. This component includes the sub-components of learning management, remote exchanges and collaboration, and online learning. The importance of this component suggests that learners value courses that are efficient and effective in delivering learning outcomes, as well as providing opportunities for collaboration and interaction with peers.

The fifth main component is “Readability”, with a defuzzied weight of 0.14 and a rank of 5. This component includes the sub-components of web-based learning, accessibility, and offline learning. The importance of this component indicates that learners prioritize having access to courses that are easy to use and navigate, as well as being accessible and available offline. The final main component is “Engagement”, with a defuzzied weight of 0.12 and a rank of 6. This component includes only one sub-component, which is the ease of use. The low importance of this component suggests that learners do not prioritize having a highly engaging course, but rather prefer courses that are easy to use and navigate.

The identified factors in e-learning, such as experts’ feedback, user adaptation, reliability of materials, course structure, methodology, format, and more, are well supported by existing literature and frameworks in e-learning and healthcare education. These factors align with instructional design models, learner-centered education principles, quality assurance frameworks, and pedagogical approaches. They draw upon established concepts like feedback from experts, personalized learning, instructional material quality, course organization, effective methodologies, multimodal delivery formats, and more. Incorporating these factors into e-learning initiatives can enhance learning outcomes and align with best practices in both e-learning and healthcare education.

## 5. Conclusions

The study followed a systematic approach, starting by identifying potential factors through a literature review and expert interviews, and then finalizing them using the nominal group technique. By distributing questionnaires and using Fuzzy ANP, the researchers were able to rank the identified factors and determine their weighted values. The main components were ranked as success, satisfaction, availability, effectiveness, readability, and engagement, with success being the most important component. The systematic approach used in this study is particularly noteworthy as it ensures that the results obtained are reliable and accurate. The first step in the process involved identifying potential factors through a thorough review of the literature and expert interviews. This step ensured that all the relevant factors were considered and that the list of identified factors was comprehensive.

The next step involved finalizing the identified factors using the nominal group technique. This technique is a structured group process that allows experts to work together to identify and prioritize the most important factors. This step ensured that the final list of factors was a consensus of expert opinions and that no important factors were overlooked.

The data were then collected by distributing questionnaires to people who deal with e-learning, and the results were analyzed using the Fuzzy ANP. This approach allowed the researchers to rank the identified factors and determine their weighted values. The pairwise comparison questionnaires distributed to 10 experts in the field helped to validate the results and ensure their accuracy.

The ranking of the main components in this study provides important insights for e-learning providers in healthcare. The finding that success is the most important component indicates that e-learning programs that are designed and delivered with a focus on success are more likely to be effective. E-learning providers should therefore ensure that their programs are designed with the goal of achieving success, and they should monitor the success of their programs to ensure that they are achieving the desired outcomes.

In conclusion, the systematic approach used in this study has provided valuable insights into the factors that are most important for the success of e-learning programs in healthcare. The findings of this study should be of interest to e-learning providers and educators who are involved in the development and delivery of e-learning programs. By focusing on the key components identified in this study, e-learning providers can improve the effectiveness of their programs and ensure that they are achieving the desired outcomes. The findings of the study have important practical implications for the development of future e-learning programs. The ranking of the main components, with success being the most important, suggests that e-learning programs should be designed with a clear focus on achieving success. E-learning providers in healthcare should prioritize success-oriented program design and delivery, regularly monitoring the success of their programs to ensure desired outcomes. Furthermore, the study recommends investigating the influence of learner characteristics, integrating emerging technologies, conducting longitudinal studies, and considering cultural and contextual variations. These recommendations can guide the development of customized and effective e-learning interventions that cater to the diverse needs of healthcare settings. By implementing these insights, e-learning providers can improve the quality and effectiveness of their programs, ultimately enhancing healthcare education and training.

In future research, it is recommended to investigate the influence of individual learner characteristics on the effectiveness of e-learning in healthcare. Furthermore, exploring the integration of emerging technologies into e-learning can enhance instructional approaches in this field. Longitudinal studies are necessary to evaluate the sustained effectiveness of e-learning programs over an extended period. However, there are some limitations of the study and highlight areas for further research. This could include acknowledging potential biases in the sample or methodology, discussing the need for additional studies to validate the findings, and suggesting new avenues of inquiry to explore unexplored aspects of e-learning in healthcare.

Additionally, investigating cultural and contextual variations will provide insights for developing customized e-learning interventions that cater to the diverse needs of healthcare settings. These avenues of inquiry will contribute to the advancement of knowledge and practice in healthcare e-learning.

## Figures and Tables

**Table 1 healthcare-11-02035-t001:** Linguistic variables’ importance description with triangular fuzzy numbers.

Linguistic Variables	Fuzzy Number	Triangular Fuzzy Number
Equally Important (EI)	1	(1, 1, 1)
Weakly More Important (WI)	3	(2/3, 1, 3/2)
Strongly More Important (SI)	5	(3/2, 2, 5/2)
Very Strongly More Important (VI)	7	(5/2, 3, 7/2)
Absolutely Important (AI)	9	(7/2, 4, 9/2)

**Table 2 healthcare-11-02035-t002:** Effective factors of e-learning extracted from literature and interviews.

Components	Sub-Components	Symbol
Success	Experts’ Feedback	C1
	User Adaption	C2
	Reliability of Materials	C3
Readability	Structure of Course	C4
	Methodology	C5
	Format	C6
Effectiveness	Planning and Training Objectives	C7
	Practice-based Learning	C8
	Materials’ Flexibility	C9
Engagement	Enjoyment and Playfulness	C10
	Learning Management	C11
	Remote Exchanges and Collaboration	C12
Availability	Online Learning	C13
	Web-based Learning	C14
	Accessibility	C15
	Offline Learning	C16
Satisfaction	Ease of Use	C17
	Reliability of Software	C18
	Overall Learner Satisfaction	C19

**Table 3 healthcare-11-02035-t003:** The normalized relation matrix.

	Success	Readability	Effectiveness	Engagement	Availability	Satisfaction
Success	0.037864	0.037864	0.037864	0.037864	0.037864	0.037864
Readability	0.019311	0.037864	0.019311	0.037864	0.013253	0.015903
Effectiveness	0.037864	0.018932	0.037864	0.018932	0.018554	0.018554
Engagement	0.019311	0.012117	0.018932	0.037864	0.016282	0.017039
Availability	0.01969	0.037864	0.019311	0.037864	0.037864	0.01401
Satisfaction	0.01969	0.037864	0.019311	0.037864	0.037864	0.037864

**Table 4 healthcare-11-02035-t004:** Factor loading of extracted variables.

Main Components	Item	Factor Loading	CR
Success	C1	0.697	0.083
	C2	0.618	
	C3	0.808	
Readability	C4	0.776	0.003
	C5	0.833	
	C6	0.636	
Effectiveness	C7	0.612	0.010
	C8	0.769	
	C9	0.637	
Engagement	C10	0.755	0.034
	C11	0.792	
	C12	0.685	
Availability	C13	0.822	0.020
	C14	0.791	
	C15	0.650	
	C16	0.639	
Satisfaction	C17	0.801	0.087
	C18	0.717	
	C19	0.713	

**Table 5 healthcare-11-02035-t005:** Fuzzy weights of main components and sub-components.

$\tilde{W_{j}}$ (Main Components)	$\tilde{w_{j}}$ (Sub-Components)	Defuzzied Weight		Rank	
$\tilde{W_{1}}$	C1	0.33	0.24	2	1
	C2	0.26		3	
	C3	0.41		1	
$\tilde{W_{2}}$	C4	0.41	0.14	1	5
	C5	0.33		2	
	C6	0.26		3	
$\tilde{W_{3}}$	C7	0.28	0.15	1	4
	C8	0.23		2	
	C9	0.18		3	
$\tilde{W_{4}}$	C10	0.25	0.12	3	6
	C11	0.41		1	
	C12	0.34		2	
$\tilde{W_{5}}$	C13	0.28	0.16	2	3
	C14	0.23		3	
	C15	0.32		1	
	C16	0.17		4	
$\tilde{W_{6}}$	C17	0.33	0.19	2	2
	C18	0.26		3	
	C19	0.41		1	

## Data Availability

Data is unavailable due to privacy.

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
