# Peer review of "Factors Influencing the Effectiveness of E-Learning in Healthcare: A Fuzzy ANP Study"

_healthcare, 2023, doi:10.3390/healthcare11142035_

Round 1

Reviewer 1 Report

Overall, the manuscript is a good paper. However some corrections and improvement are needed as follows:

1) Researcher decided to utilize Cochran's finite society sample size formula to identify adequate sample size. All those 166, 105 people who targeted to participate in the first stage of this research are dealing with 167 e-learning, mostly in healthcare.

-  cannot find Cochran's finite society sample size formula

and the calculation of sample size.

2) No explanation about the survey questionnaire development and derivation of factors. 

1) Some sentences are not properly written, for example

........[22], [23] . stressed that the transformation procedure from the industrial era to the knowledge era resulted in our current society... (see lines 73,74,75,76)

2) The formula writings need improvement.

Author Response

Dear Reviewer,                                              

We are pleased to submit the revised version of Manuscript ID healthcare-2466441 entitled "Factors Influencing the Effectiveness of E-Learning in Healthcare: A Fuzzy ANP Study". We are truly grateful for your constructive and insightful comments to guide in improving the manuscript. The changes that have been made are highlighted in the manuscript by using the track changes mode in MS Word. During the revision of the paper, some changes in the page number occurred. We have addressed each of your comments as is outlined below.

As for the comments from Reviewer #1, we have done the revision in the following way:

1) Researcher decided to utilize Cochran's finite society sample size formula to identify adequate sample size. All those 166, 105 people who targeted to participate in the first stage of this research are dealing with 167 e-learning, mostly in healthcare.

-  cannot find Cochran's finite society sample size formula and the calculation of sample size.

  • Response:
    Thanks for your comments, we have updated the writing accordingly to this comment from line 168-174.
  • 2) No explanation about the survey questionnaire development and derivation of factors.
  • Response:
    Thank you for your comment. We have changed the writing accordingly to this comment from line 163-166.
  • 3) 1) Some sentences are not properly written, for example

........[22], [23] . stressed that the transformation procedure from the industrial era to the knowledge era resulted in our current society... (see lines 73,74,75,76)

  • Response:
    Thanks for the comment. We agree with the reviewer. We revised the writing accordingly to this comment from line 73-77.

Reviewer 2 Report

Title: Factors Influencing the Effectiveness of E-Learning in Healthcare: A Fuzzy ANP Study

Introduction: The introduction provides a clear overview of the topic, highlighting the transformative nature of e-learning in healthcare education and the challenges faced by healthcare professionals in remote or rural areas. It effectively sets the context for the study and the need to identify factors that influence the effectiveness of e-learning in healthcare.

Methodology: The methodology section briefly describes the research approach used in the study, which includes a literature review, distribution of questionnaires to e-learning experts, and the application of the fuzzy Analytical Network Process (ANP) to determine the importance of selected factors. However, it would be helpful to provide more details regarding the selection of experts, the number of participants in the study, and any potential limitations or biases.

Results: The results section presents the main components influencing the effectiveness of e-learning in healthcare, ranked in order of importance: success, satisfaction, availability, effectiveness, readability, and engagement. The study identifies success as the most significant factor. It is crucial to provide specific findings from the questionnaire responses and the ANP analysis to support these rankings.

Discussion: The discussion section should expand on the findings and provide a deeper analysis of the identified factors. It should discuss how each factor contributes to the effectiveness of e-learning in healthcare and provide examples or evidence from the literature to support these claims. Additionally, it would be beneficial to compare and contrast the findings with existing studies or frameworks in the field of e-learning and healthcare education.

Implications and Applications: The article briefly mentions the benefits of e-learning in healthcare, including increased accessibility, interactivity, flexibility, knowledge management, and cost efficiency. However, it would be advantageous to provide more comprehensive discussions on how these benefits can address the challenges faced by healthcare professionals in remote or rural areas. Furthermore, practical implications and recommendations for the development of future e-learning programs should be provided, based on the study's findings.

Conclusion: The conclusion should summarize the key findings, emphasize the significance of the study, and highlight its contribution to the field of e-learning in healthcare. It could also address any limitations or areas for further research.

Overall, the article provides valuable insights into the factors influencing the effectiveness of e-learning in healthcare. To improve the article, consider incorporating the following revisions:

  1. Provide more details about the research methodology, including participant selection and potential limitations.
  2. Present specific findings from the questionnaire responses and the ANP analysis to support the rankings of the factors.
  3. Expand on the discussion of the identified factors, linking them to existing literature and frameworks in e-learning and healthcare education.
  4. Elaborate on the practical implications and recommendations for the development of future e-learning programs.
  5. Consider addressing any limitations or areas for further research in the conclusion.

By addressing these suggestions, the article will be more comprehensive, impactful, and informative for readers in the field of e-learning in healthcare.

English presentation should be improved.

Author Response

Dear Reviewer,                                              

We are pleased to submit the revised version of Manuscript ID healthcare-2466441 entitled "Factors Influencing the Effectiveness of E-Learning in Healthcare: A Fuzzy ANP Study". We are truly grateful for your constructive and insightful comments to guide in improving the manuscript. The changes that have been made in the manuscript. During the revision of the paper, some changes in the page number occurred. We have addressed each of your comments as is outlined below.

  • 1) Provide more details about the research methodology, including participant selection and potential limitations.
  • Response:
    Thanks for your comments, we have updated the writing accordingly to this comment from line 168-174.
  • 2) Present specific findings from the questionnaire responses and the ANP analysis to support the rankings of the factors.
  • Response:
    Thank you for your comment. We have changed the writing accordingly to this comment from line 163-166.
  • 3) Expand on the discussion of the identified factors, linking them to existing literature and frameworks in e-learning and healthcare education.
  • Response:
    Thanks for the comment. We agree with the reviewer. We revised the writing accordingly to this comment from line 413-421.

4) Elaborate on the practical implications and recommendations for the development of future e-learning programs.

  • Response:
  • The authors appreciate the reviewer to mentioning this point. We revised the writing accordingly to this comment from line 456-467.

5) Consider addressing any limitations or areas for further research in the conclusion.

  • Response:
  • The authors appreciate the reviewer to mentioning this point. We revised the writing accordingly to this comment from line 472-476.

Yours sincerely

Authors of the manuscript

Reviewer 3 Report

Dear Authors,

The structure of your manuscript needs to be revised and be clear with the aim to provide empirical outcomes of the literature support if you need to resubmit the study further.

The method is not understandable, ANP, AHP applied? The model regarding of the ANP is not given in the text. 

For ANP, there should be a relationship matrix provided. 

Although the manuscript shows some scientifically sound explanations, however, some areas need to be very well organized and show the clarity to address proper study design, support from the relevant literature adequately, very strong and novel methodology. The authors are not sure how they used the method,

e.g. they provide most of the time about ANP, however, after Table 2 and Table 3, they start giving the results regarding the AHP method. It is so confusing why they mixed this part which was not provided in the research flow earlier. The position of Table 2 should be situated after Table 1 because those items should be explained in an earlier stage. 

In addition, if ANP is applied, a relationship matrix should be shown in detail with equations & calculations in the tables. Some formulas provided in the method part are AHP formulas. Therefore the structure of the research is improper. The research is not solid, nor conducted correctly. 

The discussion part is also concise and not properly linked with the results and literature part. 

The introduction is very weak and does not address the primary research aim or questions to encourage the reader to focus on the paper. The main part of the healthcare literature is also very weak, and only eight references are given.

N/A

Author Response

Dear Reviewer,

Thanks for the comments. We agree with the reviewer, and we updated the article accordingly to address the comments. 

Yours sincerely

Authors of the manuscript

Reviewer 4 Report

I like your article. However, I think that it would be better if you explain all letter used in formulae before you use it. I also would exchange the order of tables 1 and 2. Table 2 should be written with three columns: first - components, second - subcomponents and third - symbol. Then you don't write it in the conclusion. Table 3 - look at explanantion of columns. 

Decide how do you name the used method: fuzzy ANP, or FANP?

I like the article; however to be better understandable, the authors should explain letters and symbols before using it. 

Author Response

We are pleased to submit the revised version of Manuscript ID healthcare-2466441 entitled "Factors Influencing the Effectiveness of E-Learning in Healthcare: A Fuzzy ANP Study". We are truly grateful for your constructive and insightful comments to guide in improving the manuscript. The changes that have been made are highlighted in the manuscript by using the track changes mode in MS Word. During the revision of the paper, some changes in the page number occurred. We have addressed each of your comments as is outlined below.

As for the comments from Reviewer #4, we have done the revision in the following way:

1) I like your article. However, I think that it would be better if you explain all letter used in formulae before you use it. I also would exchange the order of tables 1 and 2. Table 2 should be written with three columns: first - components, second - subcomponents and third - symbol. Then you don't write it in the conclusion. Table 3 - look at explanation of columns.

Decide how do you name the used method: fuzzy ANP, or FANP?

  • Response:
    Thanks for your comments, we have updated the writing accordingly to this comment, and used the same term in all of the article.
  • 2) cannot find Cochran's finite society sample size formula and the calculation of sample size.
  • Response:
    Thank you for your comment. We have changed the writing accordingly to this comment from line 167-177.
  • 3) Table 2 should be written with three columns: first - components, second - subcomponents and third - symbol.
  • Response:
    Thanks for the comment. We agree with the reviewer. We revised tables 1 and 2.

Yours sincerely

Authors of the manuscript

Round 2

Reviewer 2 Report

Dear Authors,

I would like to inform you that the review of your article has been completed, and I wanted to note that the revisions you made have brought the article to an acceptable level for acceptance.

Thank you, and best wishes for your continued success.

Author Response

Dear Reviewer,

We are truly grateful for your constructive and insightful comments to guide in improving the manuscript. 

Yours sincerely

Authors of the manuscript

Reviewer 3 Report

Dear Authors,

Thank you for your revision for the second round. Although you added some relevant material to the manuscript, your study still has significant missing parts. The study has no clear Research Questions in the introduction or Methodology part.

Only one sentence mentions, "This mixed-method research aimed to identify the effective factors for healthcare e-learning." in the Rationale section. It is not clear where is the mixed method. Do you mean that collecting data is mixed? Or have you applied at least two MCDM methods.?

The ANP model is not covered correctly (not at all), and the calculation step does not show any proof regarding the ANP, such as a supermatrix table is not provided. Fuzzy ANP steps should be applied. The consistency and appropriateness of the analysis are missing. The pairwise comparison was given; however, it is not clear. Implications or effects of criteria are not shown since the ANP requires them.

Please check the below samples for further studies or to improve your manuscript.

Uygun, Ö., Kaçamak, H., & Kahraman, Ü. A. (2015). An integrated DEMATEL and Fuzzy ANP techniques for evaluation and selection of outsourcing provider for a telecommunication company. Computers & Industrial Engineering86, 137-146. https://doi.org/10.1016/j.cie.2014.09.014

Büyüközkan, G., & Çifçi, G. (2012). A novel hybrid MCDM approach based on fuzzy DEMATEL, fuzzy ANP and fuzzy TOPSIS to evaluate green suppliers. Expert Systems with Applications39(3), 3000-3011. https://doi.org/10.1016/j.eswa.2011.08.162

The minor correction needs, in general, OK.

Author Response

Dear Reviewer,                                              

 We are truly grateful for your constructive and insightful comments to guide in improving the manuscript. The changes that have been made are highlighted in the manuscript by using the track changes mode in MS Word. During the revision of the paper, some changes in the page number occurred. We have addressed each of your comments as is outlined below.

As for the comments from Reviewer #3, we have done the revision in the following way:

  • 1) Thank you for your revision for the second round. Although you added some relevant material to the manuscript, your study still has significant missing parts. The study has no clear Research Questions in the introduction or Methodology part.
  • Response:
    Thanks for your comments, we have updated the writing accordingly to this comment in line 86-92.
  • 2) Only one sentence mentions, "This mixed-method research aimed to identify the effective factors for healthcare e-learning." in the Rationale section. It is not clear where is the mixed method. Do you mean that collecting data is mixed? Or have you applied at least two MCDM methods.?
  • Response:
    Thanks for your comments, we have updated the writing accordingly to this comment in line 169-170.
  • 3) The ANP model is not covered correctly (not at all), and the calculation step does not show any proof regarding the ANP, such as a super matrix table is not provided. Fuzzy ANP steps should be applied. The consistency and appropriateness of the analysis are missing. The pairwise comparison was given; however, it is not clear. Implications or effects of criteria are not shown since the ANP requires them.
  • Response:
    Thank you for your comment. We updated the writing accordingly to this comment and added two tables (1, 3) in results section (lines 309-322 and 388-392).

  • 4) Please check the below samples for further studies or to improve your manuscript

Uygun, Ö., Kaçamak, H., & Kahraman, Ü. A. (2015). An integrated DEMATEL and Fuzzy ANP techniques for evaluation and selection of outsourcing provider for a telecommunication company. Computers & Industrial Engineering, 86, 137-146. https://doi.org/10.1016/j.cie.2014.09.014

Büyüközkan, G., & Çifçi, G. (2012). A novel hybrid MCDM approach based on fuzzy DEMATEL, fuzzy ANP and fuzzy TOPSIS to evaluate green suppliers. Expert Systems with Applications, 39(3), 3000-3011. https://doi.org/10.1016/j.eswa.2011.08.162

  • Response:
    Thanks for the comment. We agree with the reviewer. We have changed the writing accordingly to this comment in lines 135-140.
  • 5) The structure of your manuscript needs to be revised and be clear with the aim to provide empirical outcomes of the literature support if you need to resubmit the study further.
  • Response:
    Thank you for your comment. We have changed the writing accordingly to this comment.

Yours sincerely

Authors of the manuscript